# Improvement of RNA In Situ Hybridisation for Grapevine Fruits and Ovules

**DOI:** 10.3390/ijms24010800

**Published:** 2023-01-02

**Authors:** Jin Yao, Xingmei Li, Na Wu, Songlin Zhang, Min Gao, Xiping Wang

**Affiliations:** 1State Key Laboratory of Crop Stress Biology in Arid Areas, College of Horticulture, Northwest A&F University, Xianyang 712100, China; 2Key Laboratory of Horticultural Plant Biology and Germplasm Innovation in Northwest China, Ministry of Agriculture, Northwest A&F University, Xianyang 712100, China

**Keywords:** grapevine, seedless, mRNA in situ hybridisation (ISH), ovule development, *VvHB63*, *VvTAU*

## Abstract

The European grapevine (*Vitis vinifera* L.) is one of the world’s most widely cultivated and economically important fruit crops. Seedless fruits are particularly desired for table grapes, with seedlessness resulting from stenospermocarpy being an important goal for cultivar improvement. The establishment of an RNA in situ hybridisation (ISH) system for grape berries and ovules is, therefore, important for understanding the molecular mechanisms of ovule abortion in stenospermocarpic seedless cultivars. We improved RNA in situ hybridisation procedures for developing berries and ovules by targeting two transcription factor genes, *VvHB63* and *VvTAU*, using two seeded varieties, ‘Red Globe’ and ‘Pinot Noir’, and two seedless cultivars, ‘Flame Seedless’ and ‘Thompson Seedless’. Optimisation focused on the time of proteinase K treatment, probe length, probe concentration, hybridisation temperature and post-hybridisation washing conditions. The objectives were to maximise hybridisation signals and minimise background interference, while still preserving tissue integrity. For the target genes and samples tested, the best results were obtained with a pre-hybridisation proteinase K treatment of 30 min, probe length of 150 bp and concentration of 100 ng/mL, hybridisation temperature of 50 °C, three washes with 0.2× saline sodium citrate (SSC) solution and blocking with 1% blocking reagent for 45 min during the subsequent hybridisation. The improved ISH system was used to study the spatiotemporal expression patterns of genes related to ovule development at a microscopic level.

## 1. Introduction

Clarifying gene expression patterns is important to developing a better understanding of gene function. While RNA gel blot hybridisation (northern analysis) can be used to visualise expressions of specific genes in extracts from various structures, developmental stages or conditions, it is not suitable for identifying spatiotemporal expression patterns within organs, tissues or cell types at the microscopic level [1,2]. RNA in situ hybridisation (ISH) technology is based on hybridisation between a labelled, single-strand nucleic acid probe and specific mRNA target sequences within tissues or in cells immobilised on a matrix [3]. Early uses of ISH included visualisation of gene locations on animal chromosomes [1] and the spatial localisation of specific mRNAs in tissues [4], and as a diagnostic tool for detecting virus-infected cells [5]. After 1987, ISH technology began to be applied to understanding the regulation of gene expression in plants [6]. Later, ISH was developed for many systems, with basic protocols adapted from those described by Jackson [7]. In recent years, as the most effective and direct molecular biology technique for studying the spatiotemporal distribution of gene expression at the microscopic level, RNA in situ hybridisation (ISH) has become increasingly utilised in plant studies, including in *Arabidopsis thaliana* (L.) [8], cucumber (*Cucumis sativus* L.) [9], *maize* (*Zea mays* L.) [10], *Nicotiana alata* Link et Otto [11], tomato (*Solanum lycopersicum* L.) [12], *rice* (*Oryza sativa* L.) [13] and orchid *Orchis italica* [14]. Although there are some applications of ISH with grapes [15,16,17], most of the experimental materials are focused on flowers or flesh, and immature seeds, and few methods were suited to the hard, ripe seeds of a seeded grape. Therefore, it is very important to establish an RNA in situ hybridisation system suitable for mature grape seeds and ovules.

The European grapevine (*Vitis vinifera* L.) is one of the most widely cultivated and economically important horticultural crops. Due to consumer demand for seedless cultivars for fresh fruit consumption, seedlessness has become an important goal for cultivar development of grapevines. However, there are few seedless cultivars with good fruit quality. Meanwhile, traditional breeding methods are complicated by the requirement for embryo rescue. Accordingly, there has been a groundswell of research designed to better understand the genetic and molecular bases of seedlessness in grapes. Currently, certain genes have been reported to be related to grape seed development, including the MADS-box superfamily [18,19], homeobox transcription factor family [20] and YABBY transcription factor family [21]. The ovule and developing seed are organs that comprise complex structures composed of numerous distinct cell types and include genetically distinct tissues, both maternal and embryonic. Thus, the establishment of an RNA in situ hybridisation (ISH) system to study gene expression in these structures would have great significance for advancing our knowledge of seedlessness in grapevines. While this system has already been successfully applied to the study of grapevine flowers [22,23] and fruit [24,25], there is minimal practical information available concerning the optimal conditions for use in seeds.

In a previous study, we identified two transcription factor genes, *VvHB63* and *VvTAU*, which showed differential expression in developing ovules between seedless and seeded grapevine genotypes, and thus may participate in ovule development [26]. The homeobox gene BP conserved with VvHB63 plays a key regulatory role in defining important aspects of the growth and cell differentiation of the inflorescence stem, pedicel and style in Arabidopsis [27]. Therefore, we hypothesized that the *VvHB63* might have a similar function in grapes. However, the function of *VvTau* has not yet been thoroughly studied, particularly in seed development. To further determine the expression locations of these genes, we developed an RNA in situ hybridisation (ISH) protocol for grapevine fruit (berry) and ovules, based on established ISH techniques and methods developed for *Arabidopsis thaliana* (L.) [8], cucumber (*Cucumis sativus* L.) [9] and *maize* (*Zea mays* L.) [10]. Optimisation of ISH conditions is typically required for different plants and structures because of variability in tissue composition and structure. The RNA in situ hybridisation (ISH) experimental steps are very complicated, and include the following: probe preparation, tissue section preparation, section pre-hybridisation, probe and section hybridisation, washing after hybridisation and detection and visualisation of hybridisation results. Each step in the procedure has the potential to greatly influence the final results, which depend on the probe signal, low background signal and preservation of tissue structure. We improved the overall RNA in situ hybridisation (ISH) process by improving each of these steps, and so developed a practical and efficient method for studying gene expression in the developing ovules/seeds of the European grapevine (*Vitis vinifera* L.).

## 2. Results

Except the optimization of washing conditions, other factors were optimized by using the following washing conditions: washing solution concentration: 0.1× SSC; washing times: three times; 2% blocking solution; blocking time: 45 min [28].

### 2.1. Proteinase K Treatment Time

Proteinase K (1 μg/mL) was used to optimise the digestion time during the pre-hybridisation process. As shown in Figure 1c, when the digestion time was 20 min, the hybridisation signal was weak, but when the treatment time was extended to 30 min (Figure 1d), it was much stronger and the tissue morphology remained essentially intact. Based on this, we determined that the optimum incubation time with 1 μg/mL proteinase K was 30 min. 

### 2.2. Probe Length

In order to confirm the suitable probe length for in situ hybridisation, experiments were carried out with samples treated with proteinase K for 30 min. We evaluated hybridisation with an unhydrolysed probe with a length of 402 bp and a hydrolysed probe with a length of 150 bp. We found that the use of the hydrolysed 150 bp probe resulted in a stronger signal (Figure 2d).

### 2.3. Probe Concentration

In order to obtain a feasible probe concentration, experiments were carried out with samples treated with proteinase K for 30 min and the 150 bp hydrolysed probe. Probe concentrations of 50 ng/mL, 100 ng/mL and 150 ng/mL were tested. We found that when the probe concentration was 100 ng/mL or 150 ng/mL, the hybridisation signal was sufficiently strong compared to 50 ng/mL (Figure 3d,f). However, the use of the 150 ng/mL probe resulted in a strong background signal (Figure 3f). Based on this, we concluded that the 100 ng/mL probe gave the optimal ratio of target to background.

### 2.4. Hybridisation Temperature

In order to obtain a feasible hybridisation temperature, experiments were carried out with samples treated with proteinase K for 30 min and the 150 bp hydrolysed probe at a concentration of 100 ng/mL. The hybridisation temperatures were set to 50 °C and 55 °C. When the hybridisation temperature was 50 °C, the signal was strong and the background was weak (Figure 4c); at 55 °C, the hybridisation signal was enhanced, but the background interference increased (Figure 4d). Therefore, we concluded that the feasible hybridisation temperature was 50 °C.

### 2.5. Washing Conditions

The signal-to-background ratio is strongly influenced by washing conditions after hybridisation, including the number of washings, the concentration of the washing solution and concentration of the blocking solution. We evaluated several washing conditions, under our selected conditions for proteinase K digestion time (30 min), probe length (150 bp) and concentration (100 ng/mL), and temperature (50 °C).

The results show that the hybridisation signal was weak after five serial washes with 0.1× SSC solution and blocking with 1% blocking solution for 45 min (Figure 5d). In contrast, after three washes with 0.2× SSC and sealing with 1% blocking solution for 45 min, the hybridisation signal was strong, the background signal was low and the tissue morphology was still intact (Figure 5e). When the concentration of the blocking reagent was increased to 2% following three washes with 0.2× SSC, the hybridisation signal was enhanced but the background signal increased (Figure 5f). Based on these results, we concluded that feasible conditions for washing included three washes with 0.2× SSC, followed by 1% blocking solution for 45 min.

### 2.6. Use of Improved RNA ISH Protocol for Detection of Genes Associated with Seed Development, VvTAU and VvHB63

We then applied the above in situ hybridisation conditions for analysing expression of *VvHB63* and another transcription factor (TF), *VvTAU*. Semi-quantitative analysis of the synthesised probes by dot-blot hybridisation (Appendix A) showed that the concentration of the *VvHB63* hydrolysed sense probe, hydrolysed antisense probe, unhydrolysed sense probe and unhydrolysed antisense probe was approximately 40 ng/μL and *VvTAU* probes showed that the concentration of the synthesised sense probe was ~200 ng/μL and the concentration of the antisense probe was ~40 ng/μL (Appendix A).

The in situ hybridisation analysis showed that the expression of *VvHB63* was focused on pulp cells, seed coat epidermis cells and inner integument in seedless grapes (Figure 6). At 15 days after flowering (DAF), *VvHB63* showed strong hybridisation signals in the inner integument, and epidermal cells of the ovules of ‘Thompson Seedless’ and ‘Flame Seedless’ (Figure 6a–d) and after 25 DAF, the hybridisation signals were also detected in the chalaza, middle integument, inner integument and epidermal cells of the ovules. The difference was that partial expression signals were also detected in the integument of the seeded grapes ‘Red Globe’ and ‘Pinot Noir’ (Figure 6e–l). Another transcription factor *VvTAU* has higher expression levels in seeded grapes than seedless ones (Appendix A). At 15 DAF, the integument cells of ‘Red Globe’ and ‘Pinot Noir’ showed strong expression signals for *VvTAU* (Figure 7a,c), and the signals were more intense at 25 DAF (Figure 7i,k). It could be found that almost no hybridisation signal was produced in the ovule of seedless grapes, especially in ‘Thompson Seedless’ (Figure 7b,f,j). At the same time, there were strong hybridisation signals between the epidermal cells and integument of ‘Red Globe’ and ‘Pinot Noir’ (Figure 7i,k).

## 3. Discussion

The RNA in situ hybridisation (ISH) method has been widely used to analyse gene expression in annual and biennial herbaceous plant systems. However, grapevine is a woody perennial plant and woody structures have proved to be difficult for ISH, due to their toughness and high content of reactive phenolic compounds. Although other researchers have used ripening fruits to finish RNA in situ hybridisation (ISH) experiments, this research has usually been focused on the soft tissues of ripe berries or flowers [23,25], not on the seeds. The seeds of ripe grapes are very hard and difficult to impregnate with reagents. Fluorescent in situ hybridisation (FISH) is a cytogenetic technique based on the principle of using labelled DNA probes to bind to the complementary DNA. Target DNA may be chromosomes, interphase nuclei, DNA fibres, or tissue sections, but for plants, the target DNA mainly refers to chromosomes [29]. In addition, it can also be used to visualise DNA recombination events [30]. The purpose of this study was to improve the RNA ISH protocol for tissues of the developing fruits and seeds of grapevine.

Maintenance of tissue integrity is essential if ISH signals are to be interpreted effectively. During pre-hybridisation, the tissue sections are digested with acid or proteinase to increase the accessibility of the mRNA to the tissues. Generally, lightly fixed materials require only very mild digestion, whereas more strongly fixed materials require harsher digestion. We suggest that longer proteinase K treatment times should be avoided because of their potential to cause increased non-specific probe bindings [31]. On the other hand, insufficient proteinase K treatment times will limit probe penetration of the tissue, and so reduce its binding to the mRNA [24]. In the in situ hybridisation analysis of turnips (*Brassica rapa* var. rapa), the processing time of proteinase K was 15 min and 30 min, and 30 min protease K treatment was recommended [26]. We designed a gradient with 20 min in the middle, in an attempt to shorten the treatment time of proteinase K while ensuring a clear hybridisation signal (Figure 1). In this study, fruit and ovule sections were treated with 1 μg/mL proteinase K for different times. We observed that, irrespective of the proteinase K treatment time (20 min or 30 min), the tissue morphology remained essentially intact. When the digestion time was 30 min, the hybridisation signal was stronger (Figure 1d).

During hybridisation, probe length [32], probe concentration and hybridisation temperature all affected the hybridisation results. We compared results with two probe lengths, 150 bp (formed by hydrolysation of 402 bp fragment) and the unhydrolysed probe length of 402 bp (Figure 2). We would expect relatively short probes to penetrate the tissue and bind to target RNAs more readily than long probes [25]. Although, relatively long probes contain more digoxigenin labels, which is expected to increase the intensity of the hybridisation signal, as too long a probe will lead to unclear hybridisation background and indistinct tissue structure (Figure 2a,c).

Probe concentration also affects the intensity of the hybridisation signal and the hybridisation background [25,28]. Thus, choosing an appropriate probe concentration is important to obtaining maximum signal intensity. Generally, the probe concentrations used are 100–600 ng/mL [33,34]. It has also been suggested that probe concentration should be adjusted according to probe length, with the optimal concentration being about 0.5 ng/μL/kb [35]. In this study, we trialled the following three probe concentrations: 50, 100, 150 ng/mL (Figure 3). The results show that the 100 ng/mL probe concentration generated the best hybridisation signal without an excessive background signal (Figure 3e), with the 150 ng/mL probe creating stronger hybridisation signals but with more background interference (Figure 3f) and the 50 ng/mL probe producing a weaker hybridisation signal (Figure 3d).

The hybridisation temperature has an important influence on the specificity and sensitivity of the hybridisation signal. Relatively high temperatures may increase probe binding specificity, but they may also have a negative effect on tissue morphology [36]. Considering the particular plant materials, we increased the temperature of the hybridisation to 50 °C and 55 °C (Figure 4). When the hybridisation temperature reached 55 °C, the hybridization signal was too strong and part of the tissue structure was blocked, so we suggest that 50 °C is more suitable for the in situ hybridisation experiment of grape ovules (Figure 4b,c).

During post-hybridisation, most ISH protocols involve lengthy washing procedures in SSC solutions of gradually decreasing ionic strength [37]. In this study, we compared three or five washes, two concentrations of washing solution (0.1× or 0.2× SSC), and the concentration of the blocking solution (1% or 2% blocking reagent). We obtained the best results with washing conditions of 0.2× SSC solution for three times and blocking solution with 1% blocking reagent for 45 min (Figure 5e).

To study the regulation mechanism of grape ovule abortion, we performed RNA-seq analysis and obtained a series of differentially expressed genes [26]. *VvTau* and *VvHB63* were found to have significant differences of expression between seeded and seedless grapes in the ovule development stage and we speculated they are involved in the regulation of grape ovule development. The expression level was detected in eight different grape ovule materials, including four seeded grapes and four seedless grapes, and showed that *VvHB63* was highly expressed in all stages of ovule development of seedless grapes (Appendix A), and *VvTAU* was highly expressed in the ovules of seeded grapes, especially in the beginning stage of ovule development (Appendix A). Using the optimised in situ hybridisation program to detect the expression patterns of genes related to grape ovule development, we found that *VvTAU* was expressed mainly in the epidermal cells and integument of the ovule of the seeded cultivars, while *VvHB63* was expressed mainly in the epidermal cells, middle integument and inner integument. These results provide further evidence that these two genes are involved in ovule development in grapes. These results should aid the further study of gene expression patterns within developing seeds of grapes, which is expected to contribute significantly to the knowledge of the seedlessness trait and to the development of new seedless grape cultivars.

## 4. Materials and Methods

### 4.1. Plant Materials and Sample Collection

The grapevines were maintained in the Grapevine Germplasm Collection of Northwest A&F University, Yangling, China (34°20′ N 108°24′ E) with standard cultivation management. We used two seeded cultivars, ‘Red Globe’ and ‘Pinot Noir’, and two seedless cultivars, ‘Flame Seedless’ and ‘Thompson Seedless’. For each cultivar, five healthy and vigorous plants were selected prior to flowering and inflorescences were bagged to prevent contamination by foreign pollen. Developing berries 10 days after anthesis were cut longitudinally and immediately immersed with pre-cooled formalin acetic alcohol (FAA) fixative. Seeds were dissected from berries 20 days after flowering (DAF) and fixed on ice. All samples were subjected to a vacuum treatment at 0.06–0.08 MPa for at least 30 min, until all the materials sank. The fixed samples were stored in a refrigerator at 4 °C.

### 4.2. Preparation of Transcription Template

Probes were designed for the 5′ or 3′ UTRs of the *VvHB63* and *VvTAU* genes, as annotated for a *V. vinifera* reference genome [38]. The corresponding UTR sequence was amplified from cDNA and cloned into the pGEM-T Easy vector (Promega; Madison, WI, USA); because of the potential for genetic polymorphism among the four grapevines, the cDNA from each cultivar was amplified and cloned independently. Oligonucleotide primers used in the amplifications are shown in Table 1. Appropriate restriction endonucleases were used to linearise the plasmids at restriction sites within the pGEM-T multiple cloning site region, producing 5′-extended or blunt ends. Plasmids were purified for use as transcription templates.

Probes of *VvHB63* was used to optimise the RNA ISH of grape seeds.

### 4.3. In Vitro Transcription Reaction

Digoxigenin (DIG) labelled, single-stranded RNA probes were synthesised with T7 or SP6 polymerase using the Roche DIG RNA Labelling Kit (SP6/T7) (ROCHE 11175025910, Mannheim, Germany), following the manufacturer’s instructions. The reaction was incubated at 37 °C for 2 h, then 2 μL of RNase-free DNase I was added and the reaction was incubated for an additional 20 min. Last, the reaction mixture was transferred to a 1.5 mL RNase-free tube, and 4 μL of 0.2 M EDTA was added to stop the reaction. Formulations of the reagents used are listed in Appendix A.

### 4.4. Probe Hydrolysis (Optional)

For hydrolysed probes, the pellet was resuspended in 100 μL of DEPC-H_2_O, then an equal volume of 1× carbonate buffer was added and the sample was incubated in a water bath at 60 °C. The incubation time was calculated as follows: (initial fragment length – final fragment length)/(K × initial fragment length × final fragment length), where K = 0.11 kb/min and the final fragment length was 150 bp. The hydrolysis reaction was terminated by the addition of 10 μL 10% glacial acetic acid. To recover the RNA, 0.1 mL of 3 M NaOAC (pH 5.2) and two volumes of RNase-free anhydrous ethanol were added. The sample was incubated at −20 °C for 3 h, then centrifuged at 12,000 rpm at 4 °C for 15 min. The RNA pellet was washed with 600 μL of 70% RNase-free ethanol, followed by centrifugation at 12,000 rpm at 4 °C for 7 min. The wash was repeated, and the sample was centrifuged at 12,000 rpm at 4 °C for 15 min. Last, the pellet was air-dried on ice. To resuspend the probe, 20 μL of 50% RNase-free, deionised formamide was added to the centrifuge tube that contained the precipitate, and the probe was stored at −80 °C.

### 4.5. The In Situ Hybridisation

In addition to the optimisation steps, ISH was performed with DIG-labelled probes, using a combination of the procedures outlined by Drews [39] and Jackson [7].

### 4.6. Sectioning

Embedded materials were cut to 10 µm sections with a Leica RM2265 Microtome (Wetzlar, Germany). Slides with samples with intact microstructures were selected under a light microscope and placed on a metal slide rack for subsequent hybridisation experiments.

### 4.7. Deproteinisation

Proteinase K digestion was carried out at 37 °C for 20 or 30 min. Subsequently, proteinase activity was neutralised with 0.2% glycine–PBS solution for 2 min. Lastly, samples were rinsed in 1× PBS solution for 2 min.

### 4.8. Hybridisation

The probes (402 bp or 150 bp) were diluted with 50% deionised formamide to 50 ng/mL, 100 ng/mL and 150 ng/mL, according to the results from dot-blot hybridisation. The diluted probes were incubated at 80 °C for 2 min and then immediately placed on ice. Probes were diluted with four volumes of hybridisation solution, and 150 μL of the hybridisation mixture was placed on each slide to evenly cover the sample area; cover glasses were added and slides were placed on autoclaved filter paper soaked with 2× SSC/50% formamide. Hybridisation was carried out at 50 °C or 55 °C for 14–16 h.

### 4.9. Post-Hybridisation

Following hybridisation, slides were first washed with 0.2× SSC preheated to 55 °C. The sense probe and antisense probe sections were washed separately. Slides were then washed by one of the following three methods:

Method 1: Twice at 55 °C for 1 h in 0.1× SSC solution; twice at 37° C for 5 min in 1× NTE; 37 °C for 20 min in 1× NTE containing RNase A; twice at 37 °C for 5 min in 1× NTE; three times at 55 °C for 1 h in 0.1× SSC; 1× TBS solution for 5 min; 45 min in 1% blocking reagent using a shaker.

Method 2: Twice at 55 °C for 1 h in 0.2× SSC solution; twice at 37 °C for 5 min in 1× NTE; 37 °C for 20 min in 1× NTE containing RNase A; twice at 37 °C for 5 min in 1× NTE; 55V for 1 h in 0.2× SSC; 5 min in 1× TBS solution; 45 min in 1% blocking reagent using a shaker.

Method 3: Twice at 55 °C for 1 h in 0.2× SSC solution, twice at 37 °C for 5 min in 1× NTE; 37 °C for 20 min in 1× NTE containing RNase A; twice at 37 °C for 5 min in 1× NTE; 55 °C for 1 h in 0.2× SSC; 5 min in 1× TBS solution; 45 min in 2% blocking reagent using a shaker.

### 4.10. Detection

To prevent excessive background signals, microscopic inspection was carried out periodically during the color development period. When sufficient color had developed, the slides were rinsed in TE buffer. The probe signals were detected as dark purple staining using an Olympus BX63 microscope (Olympus Corporation, Japan).

## 5. Conclusions

We identified an effective ISH protocol for analysing spatio-temporal gene expression patterns in grape berries and grape ovules. These steps may require refinement if used with different grapevine cultivars or different tissues. The conditions determined in this work are as follows: (1)Pre-hybridisation—proteinase K treatment time of 30 min;(2)Hybridisation—probe concentration of 100 ng/mL, probe length of 150 bp and temperature of 50 °C;(3)Post-hybridisation washing—three washes with 0.2× SSC solution and blocking with 1% blocking reagent for 45 min.

## Figures and Tables

**Figure 1 ijms-24-00800-f001:**
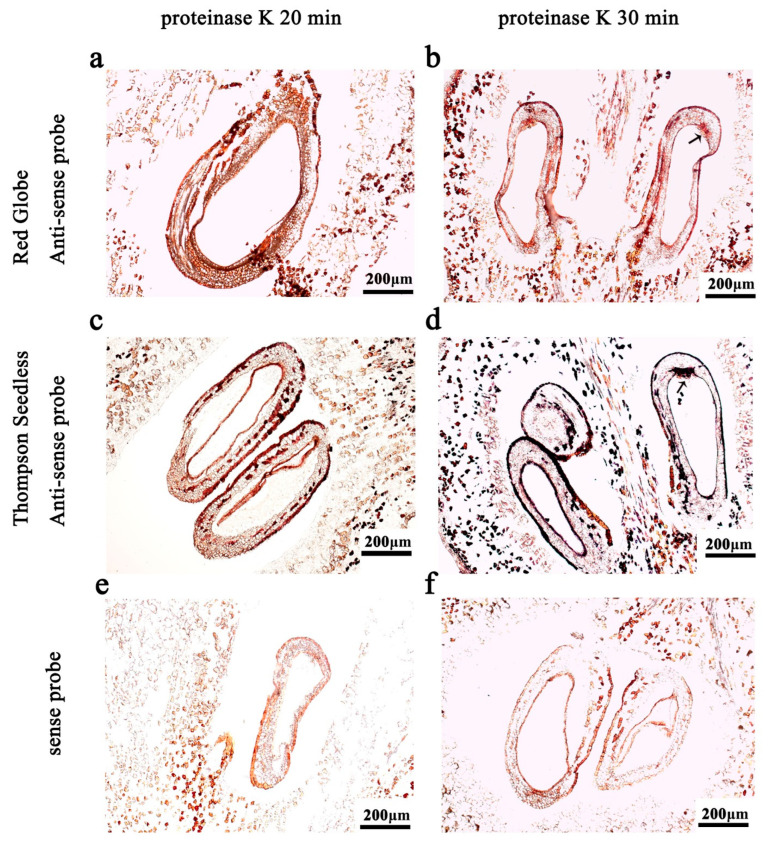
The in situ hybridisation with different digestion times of protease K: 1 µg/mL of protease K digested at 37 °C for 20 min (**a**,**c**,**e**) and 30 min (**b**,**d**,**f**). (**a**,**b**) antisense probe, 10 DAF in ‘Red Globe’; (**c**,**d**) antisense probe, 10 DAF in ‘Thompson Seedless’; (**e**,**f**) sense probe. DAF: days after flowering. The ISH signal is evidenced by purple staining and indicated by a black arrow.

**Figure 2 ijms-24-00800-f002:**
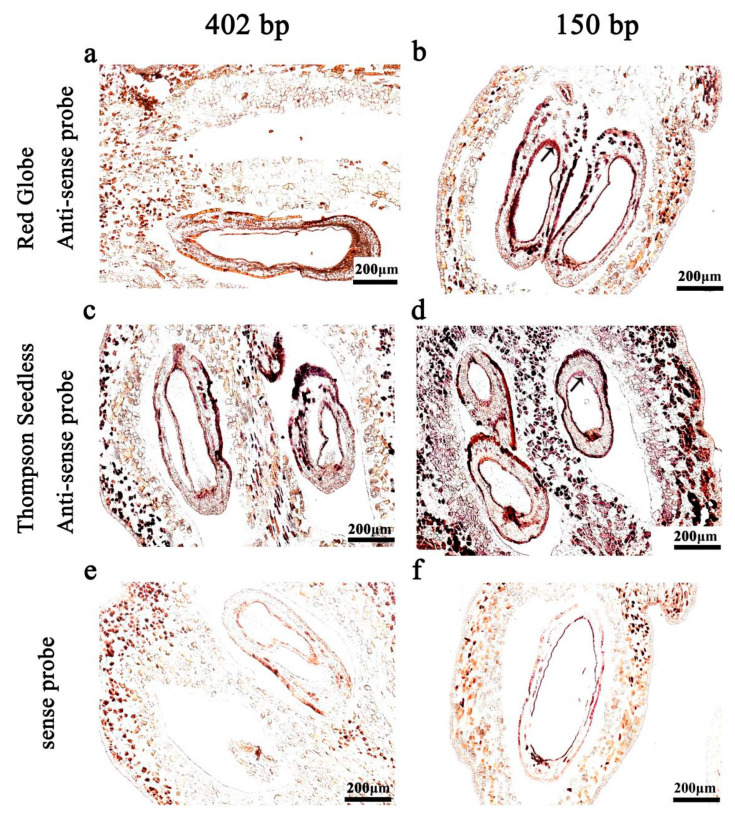
The in situ hybridisation with different probe lengths. Hybridised with the hydrolysed probe (150 bp; **a**,**c**,**e**) and unhydrolysed probe (402 bp; **b**,**d**,**f**) to produce a hybridised signal. (**a**,**b**) antisense probe, 10 DAF in ‘Red Globe’; (**c**,**d**) antisense probe, 10 DAF in ‘Thompson Seedless’; (**e**,**f**) sense probe. DAF: days after flowering. The ISH signal is evidenced by purple staining and indicated by a black arrow.

**Figure 3 ijms-24-00800-f003:**
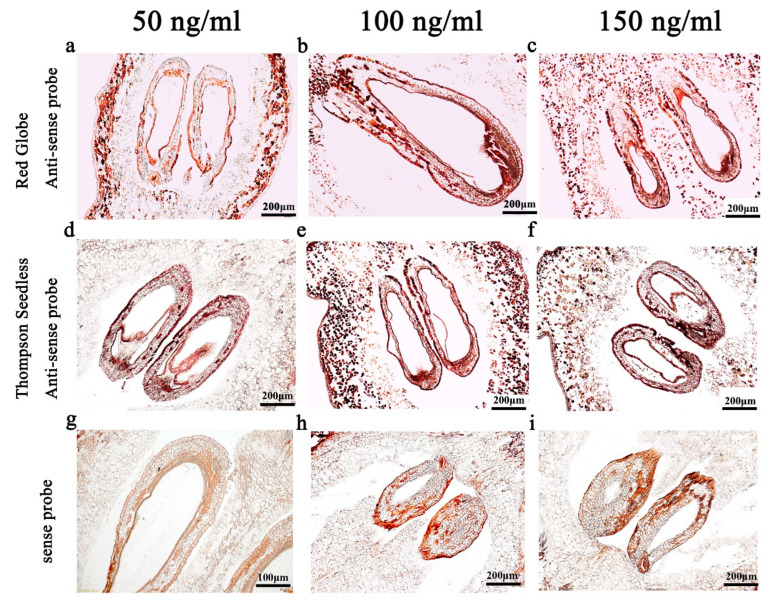
In situ hybridisation with different probe concentrations. The concentrations of the probe used were 50 ng/mL (**a**,**d**,**g**), 100 ng/mL (**b**,**e**,**h**) and 150 ng/mL (**c**,**f**,**i**). (**a**–**c**) antisense probe, 10 DAF in ‘Red Globe’; (**d**,**f**) antisense probe, 10 DAF in ‘Thompson Seedless’; (**g**–**i**) sense probe. DAF: days after flowering.

**Figure 4 ijms-24-00800-f004:**
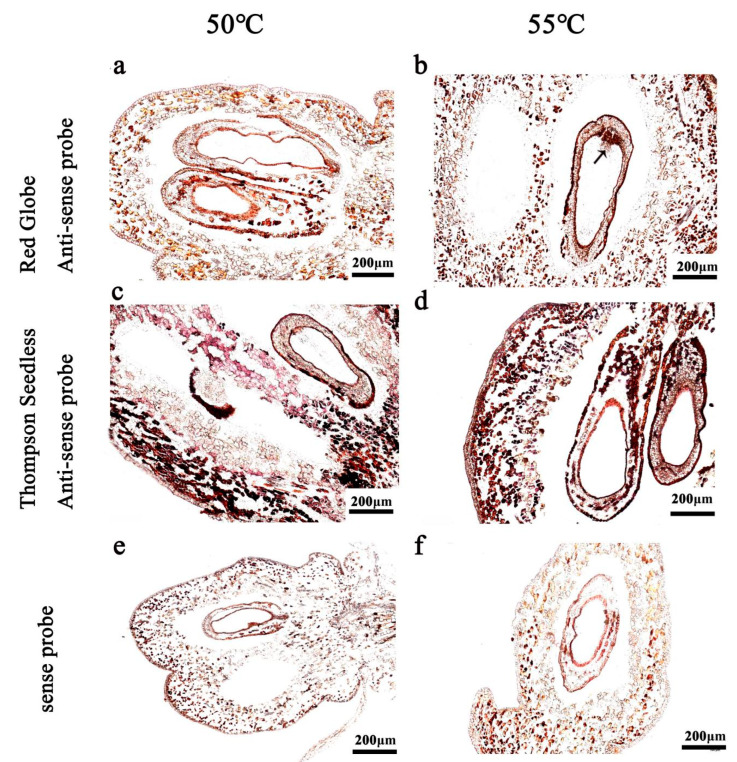
The in situ hybridisation with different hybridisation temperatures. The hybridisation temperatures were used included 50 °C (**a**,**c**,**e**) and 55 °C (**b**,**d**,**f**). (**a**,**b**) antisense probe, 10 DAF in ‘Red Globe’; (**c**,**d**) antisense probe, 10 DAF in ‘Thompson Seedless’; (**e**,**f**) sense probe. DAF: days after flowering.

**Figure 5 ijms-24-00800-f005:**
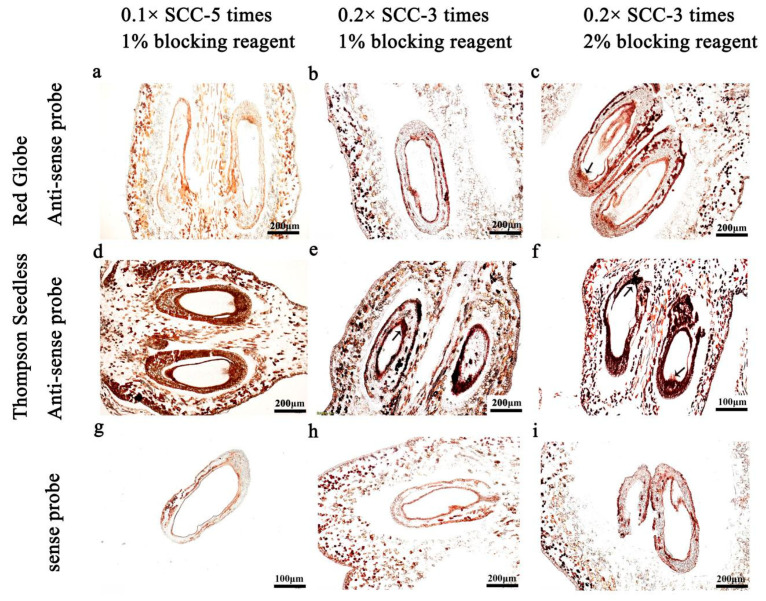
Optimisation of solution washing conditions after hybridisation. Three different washing conditions were improved by in situ hybridization, including five serial washes with 0.1× SSC solution and blocking with 1% blocking solution for 45 min (**a**,**d**,**g**), three washes with 0.2× SSC and sealing with 1% blocking solution for 45 min (**b**,**e**,**h**) and three washes with 0.2× SSC and sealing with 1% blocking solution for 45 min (**c**,**f**,**i**). (**a**–**c**) antisense probe, 10 DAF in ‘Red Globe’; (**d**,**f**) antisense probe, 10 DAF in ‘Thompson Seedless’; (**g**–**i**) sense probe. DAF: days after flowering. The ISH signal is evidenced by purple staining and indicated by a black arrow.

**Figure 6 ijms-24-00800-f006:**
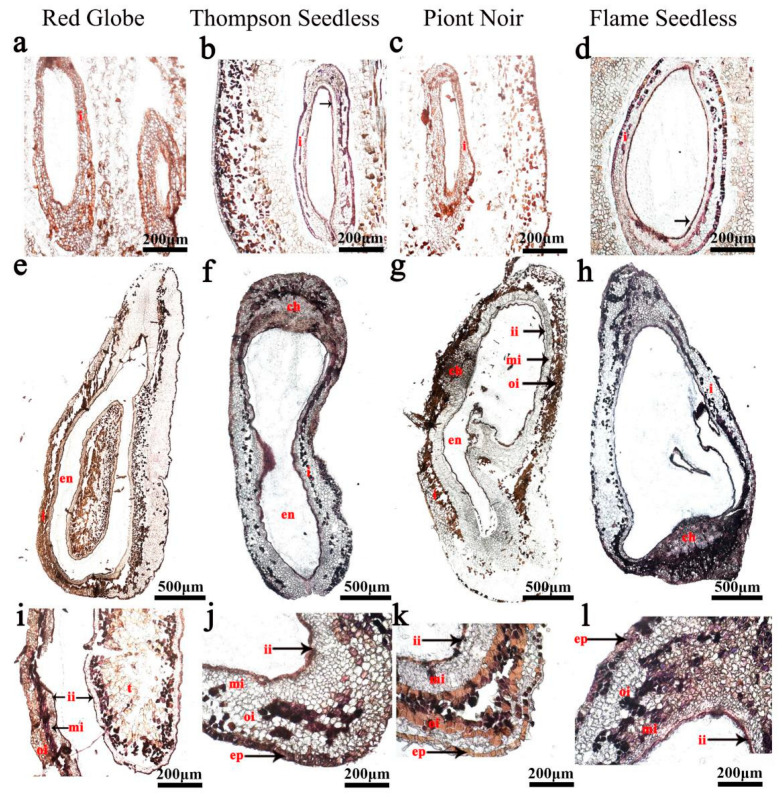
*VvHB63* expression analysis in grapevine ovules by in situ hybridisation. (**a**–**d**) 15 DAF; (**e**–**l**) 25 DAF; (i: integument; triangular label (ch): chalaza; en: endosperm; ii: inner integument; mi: medium integument; oi: outer integument; t: testa; ep: epidermis); DAF: days after flowering. At 15 DAF, *VvHB63* was expressed mainly in the epidermal cells, middle integument and inner integument (the ISH signal is evidenced by purple staining and indicated by a black arrow).

**Figure 7 ijms-24-00800-f007:**
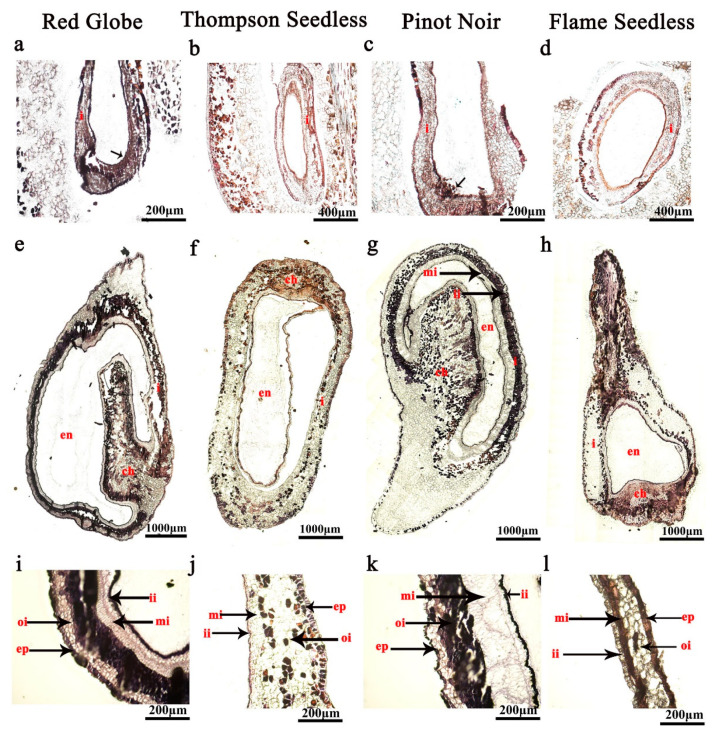
*VvTAU* expression analysis in grape ovules by in situ hybridisation. (**a**–**d**) 15 DAF; (**e**–**l**) 25 DAF; (i: integument; triangular label (ch): chalaza; en: endosperm; ii: inner integument; mi: medium integument; oi: outer integument; t: testa; ep: epidermis); DAF: days after flowering. At 15 DAF, *VvTAU* was expressed mainly in the epidermal cells and integument of the ovule of the seeded cultivars (the ISH signal is evidenced by purple staining and indicated by a black arrow).

**Table 1 ijms-24-00800-t001:** Oligonucleotide primers used for PCR amplification of transcription templates.

Gene	Forward Primer Sequence	Reverse Primer Sequence
*VvTAU*	5′ ATTTTCATGGAGTTTTTCAGCAGAT 3′	5′ TGTACAGCATTCACATCTCATCTTA 3′
*VvHB63*	5′ TGAACAAGCAGCAGGGAGGAATAG3′	5′ ATATTACACATGCTAGGAATTCCAA 3′

## Data Availability

The data that support the findings of this study are available within the article and its Appendix A.

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
