# Peer review of "Improvement of RNA In Situ Hybridisation for Grapevine Fruits and Ovules"

_ijms, 2023, doi:10.3390/ijms24010800_

Round 1
Reviewer 1 Report
Comments
The manuscript titled “Improvement of RNA in situ hybridisation for grapevine fruits and ovules” by Yao et al. presents technically informative data on RNA in situ hybridization the target tissues in grape. However, the study could have been better had the authors designed the experiments in more sound way as it lacks essential data to conclude their arguments at points. Additionally, the manuscript will benefit from a proper English editing by someone professionally proficient in English. Authors are kindly suggested to do so and requested to address the comments offered below-
Major comments
1. The authors assessed two time-point comparison for proteinase-K treatment, which showed that 30 min treatment was better than 20 min treatment. However, it may not be sufficient to conclude that any longer time treatment would essentially reduce the data quality. The preferred alternative would have been to include additional latter time-point data as well or any other suitable reference to conclude as such. Authors are kindly suggested to modify their argument (page 10; line194-195). If it cannot be concluded, it can be mentioned as study limitation.
2. Page 11; line 225-226: The manuscript does not provide enough data to conclude that VvTAU and VvHB63 are involved in ovule development. However, it inarguably suggests as such. Authors are kindly suggested to modify their argument as such.
Minor comments
1. Page 8; Figure 6: Authors are suggested to present the in-figure text in black at light background and white at dark background. It is better to point with filled arrowheads or arrows instead of unfilled triangle (in Figure 6b and 6d).
2. Authors are suggested to mention about the washing conditions used for the development of data associated with the optimum proteinase K digestion time, probe length, probe concentration, and temperature determination.
3. Authors are suggested to check and confirm whether the “Figure S1-S3” mentioned in the manuscript represent for the figures provided with those IDs in the supplementary data.
Author Response
Reviewer #1:
The manuscript titled “Improvement of RNA in situ hybridisation for grapevine fruits and ovules” by Yao et al. presents technically informative data on RNA in situ hybridization the target tissues in grape. However, the study could have been better had the authors designed the experiments in more sound way as it lacks essential data to conclude their arguments at points. Additionally, the manuscript will benefit from a proper English editing by someone professionally proficient in English. Authors are kindly suggested to do so and requested to address the comments offered below-
Response: Thank you for your suggestions. We have revised our manuscript according to your suggestions. We have also invited Dr Alexander (Sandy) Lang (JP, BSc botany, PhD plant physiology/biophysics, New Zealand) to help edit and improve our manuscript.
The authors assessed two time-point comparison for proteinase-K treatment, which showed that 30 min treatment was better than 20 min treatment. However, it may not be sufficient to conclude that any longer time treatment would essentially reduce the data quality. The preferred alternative would have been to include additional latter time-point data as well or any other suitable reference to conclude as such. Authors are kindly suggested to modify their argument (page 10; line194-195). If it cannot be concluded, it can be mentioned as study limitation.
Response: Thank you for your suggestion. We have introduced new references to support our conclusions and have re-described this part (page 11: Line 207-213).
Page 11; line 225-226: The manuscript does not provide enough data to conclude that VvTAUand VvHB63 are involved in ovule development. However, it inarguably suggests as such. Authors are kindly suggested to modify their argument as such.
Response: Thank you. In a previous study, VvHB63 and VvTAU, were identified by RNA-seq that showed differential expression in developing ovules between seedless and seeded grapevines. Further expression analysis showed that VvHB63 was highly expressed in all stages of ovule development of seedless grapes (Figure S3), and VvTau was highly expressed in the ovule of seeded grapes, especially in the beginning stage of ovule development (Figure S4). The homeobox gene AtBP-conservativeis with VvHB63 plays a key regulatory role in defining important aspects of the growth and cell differentiation of the inflorescence stem, pedicel, and style in Arabidopsis. But the functions of VvTau have not yet been studied, particularly in seed development. Therefore, we speculated that these two genes were involved in the process of grape embryo development and attempted to analyze the expression tissue sites of these two genes by in situ hybridization. We have redescribed the relevant parts (page 2: Line 69-72; page 12: Line 247-254)
Page 8; Figure 6: Authors are suggested to present the in-figure text in black at light background and white at dark background. It is better to point with filled arrowheads or arrows instead of unfilled triangle (in Figure 6b and 6d).
Response: Thank you. It has been corrected.
Authors are suggested to mention about the washing conditions used for the development of data associated with the optimum proteinase K digestion time, probe length, probe concentration, and temperature determination.
Response: Thank you for pointing this out. It has been added in a suitable place (page 3: Line 86-87).
Authors are suggested to check and confirm whether the “Figure S1-S3” mentioned in the manuscript represent for the figures provided with those IDs in the supplementary data.
Response: Thank you. It has been corrected (page 15: Line 351-360).
Reviewer 2 Report
Dear Authors,
I read this methodical manuscript with great interest, but I have some comments on it.
In the introduction, there should be information about transcription factors, and their role.
All Latin names should be in italics.
The quality of the photos is poor, and their descriptions should be expanded. It is not clear what the photo depicts.
The description of the methodology lacks information about the type, model of microscope and magnification at which the photos were taken because the size marker on the photos is completely invisible.
Please also explain why fluorescent probe marking technology was not used.
What I find missing from the discussion is a reference and comparison to other techniques that may be applicable to this type of research.
Best
M.
Author Response
Reviewer #2:
I read this methodical manuscript with great interest, but I have some comments on it.
In the introduction, there should be information about transcription factors, and their role.
Response: Thank you for pointing this out. Relevant references have been added (page 2: Line 69-72).
All Latin names should be in italics.
Response: Thank you for pointing this out. It has been corrected (page 2: Line 52).
The quality of the photos is poor, and their descriptions should be expanded. It is not clear what the photo depicts.
Response: Thank you for pointing this out. We have revised the picture and added the corresponding arrow annotations to increase understanding and added the specific picture in the results. (Fig 1-7 ).
The description of the methodology lacks information about the type, model of microscope and magnification at which the photos were taken because the size marker on the photos is completely invisible.
Response: Thank you for pointing this out. We have revised the scale of the pictures and added a the description of the micromete and microscope to the manuscript (page 14: Line 310-312; page 15: Line 338-342).
Please also explain why fluorescent probe marking technology was not used. What I find missing from the discussion is a reference and comparison to other techniques that may be applicable to this type of research.
Response: Thank you for pointing this out. By literature comparisons, we found that fluorescence probe marking technology is more suitable for chromosome localization. At the same time, the in situ hybridization technique used in this article can detect the expression site of a gene while also clearly showing the morphology of the surrounding tissues. We have added relevant references and comparisons in the Discussion (page 11: Line 198-202).
Reviewer 3 Report
The overall manuscript is written well and new knowledge is reported in.
Some suggestions to improve it:
Individual optimized parameters of in situ hybridization shoul be discussed more deeply - if they correspond to other reported ...
The common used parts in Material and Methods should be cited.
Typos and formatting should be checked. Lines 48 - 51 is hard to understand.
Figures shoul be marked more visible.
Author Response
Reviewer #3:
The overall manuscript is written well and new knowledge is reported in. Some suggestions to improve it:
Individual optimized parameters of in situ hybridization should be discussed more deeply - if they correspond to other reported ...
Response: Thank you for pointing this out. We have revised the Discussion and made references where appropriate (page 11: Line 207-213; page 11: Line 222-224; Line 230-233; Line 236-240)
The common used parts in Material and Methods should be cited.
Response: Thank you for pointing this out. We have revised the description of the methods and added the appropriate reference. (page 14: Line 310-312)
Typos and formatting should be checked. Lines 48 - 51 is hard to understand.
Response: Thank you for pointing this out. It has been corrected (page 2: Line 47-51).
Figures should be marked more visible.
Response: Thank you. We have revised the scale for the pictures.